# When and why randomised exploration works (in linear bandits)

**Marc Abeille**                                                                 M.ABEILLE@CRITEO.COM
*Criteo AI Lab, Paris, France*

**David Janz**                                                                   DAVID.JANZ93@GMAIL.COM
*University of Oxford, UK*

**Ciara Pike-Burke**                                                       C.PIKE-BURKE@IMPERIAL.AC.UK
*Imperial College London, UK*

**Editors:** Gautam Kamath and Po-Ling Loh

## Abstract

We provide an approach for the analysis of randomised exploration algorithms like Thompson sampling that does not rely on forced optimism or posterior inflation. With this, we demonstrate that in the $d$-dimensional linear bandit setting, when the action space is smooth and strongly convex, randomised exploration algorithms enjoy an $n$-step regret bound of the order $O(d\sqrt{n}\log(n))$. Notably, this shows for the first time that there exist non-trivial linear bandit settings where Thompson sampling can achieve optimal dimension dependence in the regret.

**Keywords:** linear bandits, randomised exploration, Thompson sampling

## 1. Introduction

To achieve low regret in sequential decision-making problems, it is necessary to balance exploration (selecting uncertain actions) and exploitation (selecting previously successful actions). One method of balancing this exploration-vs-exploitation trade-off that is particularly well-understood is through optimism: optimistic algorithms maintain a set of statistically plausible models of the environment and select actions that maximize the reward in the best plausible model—note however that this entails solving a bi-level optimization problem in each round. Randomised exploration is an alternative approach where algorithms select a model of the problem randomly from a set of plausible models and act optimally with respect to that randomly sampled model—bypassing the need to solve the bi-level optimization problem associated with optimism. Notable examples of randomised decision-making algorithms include Bayesian algorithms such as posterior sampling (Thompson, 1933, also known as Thompson sampling), ensemble sampling (Lu and Van Roy, 2017; Janz et al., 2024a) and perturbed history exploration (Kveton et al., 2020; Janz et al., 2024b). However, while randomisation-based algorithms are often preferred in practice, our theoretical understanding of when and why randomised exploration works in structured sequential decision-making problems is limited.

   In this paper, we analyse randomised sequential decision-making algorithms in the classic linear bandit problem—but the techniques that we introduce should carry over to other structured settings. In this setting, previous frequentist analyses (e.g. Agrawal and Goyal, 2013; Abeille and Lazaric, 2017; Kveton et al., 2020; Janz et al., 2024b) are not sufficient to explain the practical effectiveness of randomised exploration, nor do they identify a mechanism through which randomised exploration works. Indeed, existing proofs rely on modifying randomised exploration algorithms so that they can be analysed using the optimism framework. These modifications often lead to suboptimal regret. Our analysis does away with such modifications; it holds under the assumption that the action space is

smooth and strongly convex (see Section 4.2 for formal definitions), which allows for perturbation in the model parameter space to translate to perturbations in the action space, while also guaranteeing that small changes in the action space only lead to small changes in the incurred regret.

For such smooth, strongly convex action sets, which include $\ell_p$-balls for $p \in (1, \infty)$, we a regret of the order $O(d\sqrt{n}\log(n))$ where $d$ is the dimension of the action space and $n$ is the number of rounds. Notably, this shows for the first time that (unmodified) linear Thompson sampling can enjoy regret with the optimal dependence on the dimension in a structured linear bandit settings, thus partially resolving an important open question (Russo et al., 2018).

## 2. Related work

Lower bounds for the linear bandit problem depend on the structure of specific action spaces (for example, Dani et al., 2008; Rusmevichientong and Tsitsiklis, 2010; Lattimore and Szepesvari, 2017). Theorem 2.1 of Rusmevichientong and Tsitsiklis (2010) shows that there exists a problem instance where the action space is the $d$-dimensional unit sphere in which any policy must incur $\Omega(d\sqrt{n})$ regret. Optimistic algorithms have frequentist regret nearly matching the lower bound for linear bandits (Auer, 2002; Dani et al., 2008; Abbasi-Yadkori et al., 2011). Specifically, Abbasi-Yadkori et al. (2011) show that by constructing confidence sets using self-normalized bounds for vector-valued martingales, and taking actions optimistically within these, the resulting regret is $O(d\sqrt{n}\log(n/\delta))$ with probability at least $1 - \delta$. Despite the strong theoretical performance of optimistic algorithms, randomised algorithms, such as Thompson sampling, have been shown to perform better in practice (Chapelle and Li, 2011; May et al., 2012). In the simpler multi-armed bandit setting, randomised algorithms achieve optimal regret (Agrawal and Goyal, 2012; Kaufmann et al., 2012; Korda et al., 2013; Honda and Takemura, 2014). Under Bayesian assumptions, where regret is defined by taking an expectation over the unknown parameter, Russo and Van Roy (2014, 2016) show that Thompson sampling is near-optimal in many structured and unstructured settings. In particular, for the linear bandit setting, they show a Bayesian regret bound of $\widetilde{O}(d\sqrt{n})$ (Russo and Van Roy, 2014).

In this paper, our focus is on the regret of randomised exploration algorithms in linear bandits. While this setting has been studied extensively by, amongst others, previous approaches rely on modifying the algorithm to force it to be more optimistic. The main line of analysis, by Agrawal and Goyal (2013), Abeille and Lazaric (2017), and Xu et al. (2023), inflates the variance of the posterior over models in round $t$ by a factor of $\Theta(\sqrt{d\log(t/\delta)})$ to show that the algorithm is optimistic with constant probability—this leads $O((d\log(n))^{3/2}\sqrt{n})$ regret, where the increased dependence on $d$ is due to the inflation of the posterior. Further variants of randomised exploration algorithms include modifying the algorithms to only sample parameters with reward greater than the mean (May et al., 2012; Vaswani et al., 2020) and modifying the likelihood used in the Bayesian update of Thompson sampling to force the algorithm to be more optimistic (Zhang, 2022; Huix et al., 2023). The analysis of Thompson sampling in other structured settings, such as generalised linear bandits, relies on these same modifications Kveton et al. (2020) and Janz et al. (2024b).

We remark that the results presented in this paper do not contradict the lower bounds by Hamidi and Bayati (2023) and Zhang (2022) where examples were provided for which linear Thompson sampling incurs linear regret if the posterior distribution is not inflated. The action spaces constructed in those examples fail to satisfy our assumptions.

## 3. Problem setting, notation and basic definitions

We study the linear bandit problem, where each bandit instance is parameterised by an unknown $\theta_\star \in R\mathbb{B}_2^d$ ($R > 0$ known), and an action set $\mathcal{X}$, a closed subset of $\mathbb{B}_2^d$ (the closed unit $\ell_2$-ball in $\mathbb{R}^d$). Then, at each time-step $t = 1, 2, \ldots$ an agent selects an action $X_t \in \mathcal{X}$, allowed to depend on observations from previous time-steps, and receives a real-valued reward $Y_t$. We assume that the reward $Y_t$ is $S$-subgaussian given $X_t$ and the past ($S > 0$ known), with mean given by $\langle X_t, \theta_\star \rangle$. The goal of the agent is to select actions to minimize the $n$-step regret ($n \geq 1$), defined by

$$R_n = \sum_{t=1}^{n} r_t \quad \text{for} \quad r_t = \langle x_\star - X_t, \theta_\star \rangle,$$

where $x_\star \in \arg\max_{x \in \mathcal{X}} \langle x, \theta_\star \rangle$ is any optimal arm and the horizon $n$ needs not be known.

**Confidence set construction**   The algorithms and analysis in this work are based on the standard regularised least-squares-based confidence ellipsoids for $\theta_\star$ (Abbasi-Yadkori et al., 2011). To construct these, fix a regularisation parameter $\lambda > 0$ and a confidence parameter $\delta \in (0, 1)$. Define the regularised design matrices and least-squares estimates as $V_0 = \lambda I$, $\hat{\theta}_0 = 0$ and then

$$V_t = X_t X_t^\mathsf{T} + V_{t-1} \quad \text{and} \quad \hat{\theta}_t = V_t^{-1} \sum_{i=1}^{t} Y_i X_i \qquad \text{for} \quad t \geq 1,$$

Also, define the sequence of nondecreasing, nonnegative confidence widths

$$\beta_t = R\sqrt{\lambda} + S\sqrt{2\log(1/\delta) + \log(\det(V_t)/\lambda^d)}, \quad t \geq 0.$$

Then, Abbasi-Yadkori et al. (2011) show that, with probability $1 - \delta$, $\theta_\star \in \cap_{t \geq 1} \Theta_{t-1}$ for the ellipsoids given by

$$\Theta_{t-1} = \{\theta \in \mathbb{R}^d : \|\theta - \hat{\theta}_{t-1}\|_{V_{t-1}} \leq \beta_{t-1}\}, \quad t \geq 1,$$

where for $a \in \mathbb{R}^d$ and a $d \times d$ positive-definite matrix $B$, we denote by $\|a\|_B$ the $B$-weighted Euclidean norm of $a$ given by $\sqrt{\langle Ba, a \rangle}$.

**Optimistic algorithms**   Optimistic algorithms select actions $X_t$ by solving the bi-level optimization problem $(X_t, \theta_t) \in \arg\max_{(x,\theta) \in \mathcal{X} \times \Theta_{t-1}} \langle x, \theta \rangle$ in each round $t \leq n$. We instead consider randomised algorithms which randomise over $\Theta_{t-1}$. These methods are formally defined in Section 4.1

**Bregman divergence**   Our analysis will make use of a generalised Bregman divergence, defined convex function $f : \mathbb{R}^d \to \mathbb{R}$ as

$$D_f(x, y) = f(x) - f(y) - \langle \nabla f(y), x - y \rangle,$$

for almost every $y \in \mathbb{R}^d$, where $\nabla f$ denotes the gradient of $f$. We recall that convex functions are almost everywhere differentiable (Rockafellar, 1970, Theorem 25.5).

**Probabilistic formalism**   Let $\mathbb{F} = (\mathbb{F}_t)_{t \geq 0}$ be a filtration where $\mathbb{F}_0$ is the trivial $\sigma$-algebra and $\mathbb{F}_t = \sigma(\sigma(X_t, Y_t), \mathbb{A}_t)$, where $\mathbb{A}_t$ is the $\sigma$-algebra generated by any additional random variables the algorithm uses in selecting $X_t$. Note that this means that $X_t$ is $\mathbb{F}_t$-measurable. We will write $\mathbb{P}_t$ for the $\mathbb{F}_t$-conditional probability measure and $\mathbb{E}_t$ for the corresponding expectation. With this, we formalise the assumption that for all $t \geq 1$, $Y_t$ is conditionally $S$-subgaussian as that

$$\mathbb{E}_{t-1} \exp\{sY_t\} \leq \exp\{s^2 S^2/2\} \quad \text{for all} \quad s \in \mathbb{R}, \, t \geq 1. \tag{1}$$

**Asymptotic notation**   We will write $f(x) \lesssim g(x)$ if $f(x) = O(g(x))$, and use $\gtrsim$ for the converse.

**Vectors, norms, balls & spheres**   We will write $\| \cdot \|$ to denote the $\ell_2$-norm. We recall that for a positive-definite matrix $B$ and a vector $a$ of compatible dimensions, $\|a\|_B = \sqrt{\langle Ba, a\rangle}$ denotes the $B$-weighted $\ell_2$ norm. We write $\mathbb{B}_2^d$ for the closed unit Euclidean ball in $\mathbb{R}^d$, and $\mathbb{S}_2^{d-1}$ for its surface $\partial \mathbb{B}_2^d$, the $(d-1)$-sphere.

## 4. A frequentist regret bound for randomised algorithms in linear bandits

In this section, we state our main result that provides conditions under which randomised exploration algorithms can achieve frequentist regret of $\widetilde{O}(d\sqrt{n})$ in the linear bandit setting. We begin by describing the algorithmic framework and assumptions for the action set under which it holds.

### 4.1. Randomised algorithms: definition and assumptions

We consider algorithms that at each time-step $t \geq 1$ sample a parameter of the form

$$\theta_t = \hat{\theta}_{t-1} + V_{t-1}^{-1/2}\eta_t \,,$$

where $(\eta_t)_{t \geq 1}$ is a sequence of independent random variables (perturbations), and select action

$$X_t \in \arg\max_{x \in \mathcal{X}}\langle x, \theta_t\rangle \,.$$

Our result will require the following assumptions to hold for the perturbations $(\eta_t)_{t \geq 1}$.

**Assumption 1.** *The perturbations* $(\eta_t)_{t \geq 1}$ *are independent rotationally-invariant random variables for which there exists a constant $K > 0$ such that*

$$1 \leq \mathbb{E}\langle u, \eta_t\rangle^2 \leq K^2 \quad and \quad \mathbb{E}\langle u, \eta_t\rangle^4 \leq K^4 \quad for\ all \quad u \in \mathbb{S}_2^{d-1}\,, \ t \geq 1\,.$$

These assumptions hold for many common distributions, such as standard Gaussian (with $K^4 = 3$), and the uniform distribution on $\sqrt{d}\mathbb{S}_2^{d-1}$ (with $K = 1$).

### 4.2. Action set assumptions: smoothness and strong convexity

A core part of our contribution is in identifying the properties of action sets that allow randomised exploration to succeed. Our assumptions will be expressed in terms of the support function of $\mathcal{X}$,

$$J_{\mathcal{X}}(\theta) = \max_{x \in \mathcal{X}}\langle x, \theta\rangle \,.$$

Crucially, for randomised algorithms where for each $t \geq 1$, the $\mathbb{F}_{t-1}$-conditional law of $\theta_t$ is diffuse (implied by rotational invariance), we have that

$$X_t = \nabla J_{\mathcal{X}}(\theta_t) \quad \text{almost surely for all } t \geq 1 \,.$$

Our upcoming assumptions ensure that $\nabla J_{\mathcal{X}}$ is a suitably regular function. Note that the above relation means the per-step regret of randomised algorithms is given by the divergence

$$r_t = J_{\mathcal{X}}(\theta_\star) - \langle X_t, \theta_\star\rangle = J_{\mathcal{X}}(\theta_\star) - J_{\mathcal{X}}(\theta_t) - \langle \nabla J_{\mathcal{X}}(\theta_t), \theta_\star - \theta_t\rangle = D_{J_{\mathcal{X}}}(\theta_\star, \theta_t) \,,$$

again, almost surely with respect to the $\mathbb{F}_{t-1}$-conditional law of $\theta_t$.

Our assumptions will be based on the following three definitions:

**Definition 1** (Absorbing set). *We call a set $\mathcal{X} \subset \mathbb{R}^d$ absorbing if it is a neighbourhood of the origin.*

**Definition 2** (Strong convexity). *We say $J_{\mathcal{X}}^2$ is $m$-strongly convex with respect to a norm $\| \cdot \|_*$ if*

$$\frac{m}{2}\|\theta - \theta'\|_*^2 \leq D_{J_{\mathcal{X}}^2}(\theta, \theta') \quad \text{for all} \quad \theta, \theta' \in \mathbb{R}^d .$$

**Definition 3** (Smoothness). *We say that $J_{\mathcal{X}}^2$ is $M$-smooth with respect to a norm $\| \cdot \|_*$ if*

$$D_{J_{\mathcal{X}}^2}(\theta, \theta') \leq \frac{M}{2}\|\theta - \theta'\|_*^2 \quad \text{for all} \quad \theta, \theta' \in \mathbb{R}^d .$$

With these definitions in place, the conditions we will ask for on the arm set $\mathcal{X}$ are captured thus.

**Assumption 2.** *The action set $\mathcal{X}$ is a closed absorbing subset of $\mathbb{B}_2^d$, and there exists a norm $\| \cdot \|_*$ and constants $M, m > 0$ such that $J_{\mathcal{X}}^2$ is $m$-strongly convex and $M$-smooth.*

The motivation for asking for strong convexity and smoothness for the square $J_{\mathcal{X}}^2$, rather than directly for $J_{\mathcal{X}}$, is that the quantity

$$\nabla J_{\mathcal{X}}^2(\theta) = 2J(\theta)\nabla J(\theta) \tag{2}$$

does not explode as $\theta \to 0$, whereas $\nabla J_{\mathcal{X}}(\theta)$ does. That $\mathcal{X}$ is absorbing ensures that the multiplier $J(\theta)$ in the above is positive, which will come in useful in our proofs—we do not believe this assumption to be essential, but we have thus far been unable to eliminate it.

**Remark 1.** *Definition 3 generalises the notion of $M$-strong convexity used in Rusmevichientong and Tsitsiklis (2010), where this was defined by the requirement that*

$$\|\nabla J_{\mathcal{X}}(\theta) - \nabla J_{\mathcal{X}}(\theta')\| \leq M\|\theta - \theta'\| \quad \text{for all} \quad \theta, \theta' \in \mathbb{S}_2^{d-1} .$$

*Our definition will be vital to getting the right rate for randomised algorithms outside the $\ell_2$-ball case, and specifically to avoid incurring an extra factor of $\|\theta_\star\|/J(\theta_\star)$ in the regret, which may be large. We note also that their definition is for the strong convexity of the arm-set, whereas our definition is for the smoothness of $J_{\mathcal{X}}^2$. There is a duality between the (indicator function of) the set and the corresponding support function, which explains the inversion in the nomenclature.*

**Remark 2.** *If $\mathcal{X}$ is absorbing and balanced (symmetric about the origin), $J_{\mathcal{X}}$ is a norm; if it is just absorbing, $\tilde{J}(\theta) = J_{\mathcal{X}}(\theta) \vee J_{\mathcal{X}}(-\theta)$ is a norm. In these cases, it may be productive to try taking $\| \cdot \|_* = J_{\mathcal{X}}(\cdot)$ (or $\tilde{J}(\cdot)$), as in our above examples. Of course, $\| \cdot \|_*, m, M$ do not need to be known to run the algorithm, and the regret implicitly scales with the best $M/m$ over all norms $\| \cdot \|_*$.*

An example of an action sets that satisfy Assumption 2 are $\ell_p$ balls with $p \in (1, \infty)$:

**Example 1.** *Let $p, q > 1$ be conjugate indices ($\frac{1}{p} + \frac{1}{q} = 1$), $\mathcal{X} = \mathbb{B}_q^d$ and $\| \cdot \|_* = \| \cdot \|_p$. Then, Assumption 2 holds with $m = 1$, $M = (p-1)$ for $q \in (1, 2)$ and $m = p - 1$, $M = 1$ for $q \in [2, \infty)$.*

Assumption 2 is unaffected by linear transformations, extending the above examples to ellipsoids:

**Example 2.** *Let $\mathcal{X}$ be any arm set satisfying Assumption 2 for some $\| \cdot \|_*$, $M$ and $m$. Then, for any $A \in \mathbb{R}^{d \times d}$, $A\mathcal{X} := \{x \in \mathbb{R}^d : Ax \in \mathcal{X}\}$ satisfies Assumption 2 for norm $x \mapsto \|Ax\|_*$, $M$ and $m$.*

### 4.3. Main result and discussion

We are now ready to state our main result which shows that any randomised algorithm satisfying Assumption 1 with an action set satisfying Assumption 2 achieves at most $\widetilde{O}(d\sqrt{n})$ regret in the linear bandit problem. This matches the lower bound of Rusmevichientong and Tsitsiklis (2010) up to logarithmic factors (based on $\mathcal{X} = \mathbb{B}_2^d$, a set that satisfies our assumptions).

**Theorem 4.** *Fix $\lambda \geq 1$ and $\delta \in (0,1)$. Suppose that a learner uses a randomised algorithm with perturbations satisfying Assumption 1 on a linear bandit instance with an arm-set that satisfies Assumption 2. Then, for any $\theta_\star \in \sqrt{d}\mathbb{B}_2^d$, with probability $1 - \delta$, for all $n \geq 1$, the $n$-step regret incurred by the learner is bounded as*

$$R_n \lesssim \frac{M}{m} K(\beta_n^2 \vee K^2 d)\sqrt{n} + K^4 \beta_n \sqrt{n(d\log(1 + n/(d\lambda)) + \log(1/\delta))},$$

The proof of this result is presented in Section 5, with much of the details deferred to the appendices. We now discuss some aspects of our result, its proof and its relation to previous works.

**On the regret of Thompson sampling**    If the noise in the responses $(Y_t)_{t \geq 1}$ is Gaussian with a known variance $\sigma^2$, and if for all $t \geq 1$ the perturbations are given by $\eta_t \sim \mathcal{N}_d(0, \sigma^2 I)$, then our randomised exploration algorithm is equivalent to the linear Thompson sampling algorithms of Russo and Van Roy (2014), Agrawal and Goyal (2012), and Abeille and Lazaric (2017). Thus, for action spaces satisfying Assumption 2, Theorem 4 shows that Thompson sampling can enjoy regret of $O(d\sqrt{n}\log(n))$, leaving at most an $O(\log n)$ gap between this frequentist regret and the corresponding Bayesian regret (see Russo and Van Roy, 2014, 2016, for Bayesian analyses).

**On the lower bound for randomised algorithms**    We remark that Theorem 4 holds for any randomised algorithm without any modification; in particular there is no need to inflate any variance proxies. This is in contrast to lower bounds by Hamidi and Bayati (2023) and Zhang (2022) which show that there exist problem instances on which linear Thompson sampling suffers linear regret. These instances are specifically designed so that there is a bad 'trap' arm, where pulling that arm yields regret, but no information, so that Thompson sampling gets stuck. They are the polar opposite of what Assumption 2 asks for: not absorbing, strongly convex, or smooth.

**Limitation of optimism-based proofs**    Existing proofs of frequentist regret bounds for randomised algorithms in linear bandit, including those of Agrawal and Goyal (2013), Abeille and Lazaric (2017), Kveton et al. (2020), and Janz et al. (2024b), leverage that with high probability,

$$r_t = J_{\mathcal{X}}(\theta_\star) - \langle X_t, \theta_\star \rangle = D_{J_{\mathcal{X}}}(\theta_\star, \theta_t) \leq \sup_{\theta \in \Theta_t} D_{J_{\mathcal{X}}}(\theta_\star, \theta),$$

and then show that $\sup_{\theta \in \Theta_t} D_{J_{\mathcal{X}}}(\theta_\star, \theta)$ can be suitably controlled when randomised sampling guarantees sufficient optimism—that is, when the algorithm is optimistic with a fixed probability. Unfortunately, as illustrated in Abeille and Lazaric (2017, Fig. 2), guaranteeing optimism with a fixed probability requires inflating the variance of the sampling distributions, and this results in an extra $\sqrt{d}$ factor in the regret bound. Moreover, these proofs implicitly suggest that non-optimistic samples do not help in controlling the upper bound on the per-step regret, $\sup_{\theta \in \Theta_t} D_{J_{\mathcal{X}}}(\theta_\star, \theta)$.

    This approach is overly conservative in two ways: first, while a particular sample may provide very little information—measured through the design matrix update $X_t X_t^\mathsf{T} = V_t - V_{t-1}$—the

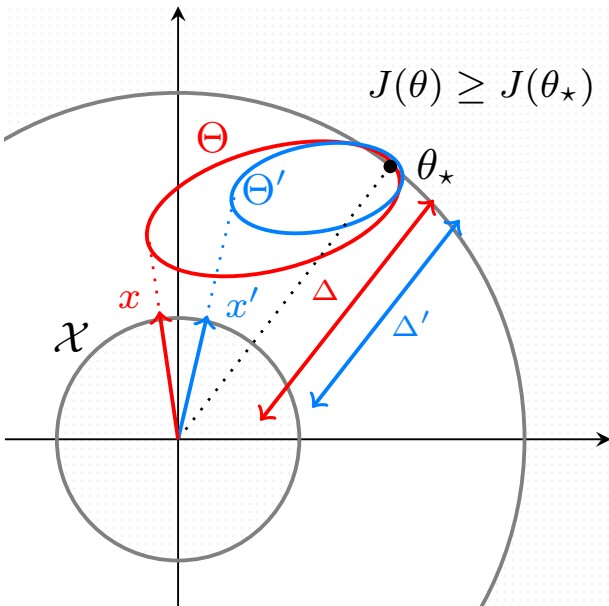

Figure 1: Illustration of the update to the confidence sets during non-optimistic exploration, and the impact this has on the per-step worst case regret, when $\mathcal{X} = \mathbb{B}_2^d$. In red, we have an initial confidence set $\Theta$; the corresponding worst-case optimal action over $\Theta$ is given by $x = \arg\min_{\theta \in \Theta} \langle \nabla(\theta), \theta_\star \rangle$ and the associated per-step worst case regret is $\Delta = \|\theta_\star\|_2 - \langle x, \theta_\star \rangle$. In blue, we illustrate the average of the respective quantities after randomised structured exploration with $\theta \sim \Theta$. That is, taking $V' = V + \mathbb{E}_{\theta \sim \Theta}(\nabla(\theta)\nabla(\theta)^\mathsf{T})$. While the actions sampled by this strategy are unlikely to be optimistic, this randomised strategy does in fact explore—the confidence set shrinks—and this reduces the per-step regret.

sample may still provide useful information *on average*, that is, by considering $\mathbb{E}_{t-1} X_t X_t^\mathsf{T}$. Second, while the information acquired at a time $t$ might not significantly reduce the per-step regret bound $\sup_{\theta \in \Theta_{t+1}} D_{J_\mathcal{X}}(\theta_\star, \theta)$ for the step immediately following it, it may prove useful at later steps. Figure. 1 illustrates how non-optimistic samples provide useful information that is ignored by the optimistic proof approaches.

**Our proof techniques**   The key challenge in developing a non-optimistic proof for randomised algorithms in linear bandits is to directly analyse the *dynamic of the exploration*, that is, of the process $\{\Theta_t\}_{t \geq 0}$, and relating this to the upper bound of the per-step regret process, $\sup_{\theta', \theta \in \Theta_t} D_{J_\mathcal{X}}(\theta', \theta)$. Interestingly, such approach is closer to the analysis of Thompson sampling in the $K$-armed bandit setting, for which it is shown to be optimal (Kaufmann et al., 2012; Agrawal and Goyal, 2012). Within the proof of our regret bound, Theorem 4, we address the above points by:

(i) Providing a new bound on $\sup_{\theta', \theta \in \Theta_t} D_{J_\mathcal{X}}(\theta', \theta)$, $t \geq 0$ by leveraging strong convexity and smoothness;

(ii) Characterising the minimum amount of information acquired during interaction through a lower bound on $V_t$, where $V_t$ acts as a proxy for $\Theta_t$;

and connecting (i) and (ii) by studying the properties of the *average per-step* information $\mathbb{E}_{t-1} X_t X_t^\mathsf{T}$.

**Comparison with forced exploration**  Rusmevichientong and Tsitsiklis (2010) proposes a phased explore-then-commit algorithm that interleaves rounds of playing $d$ linearly independent actions with increasingly long exploitation phases, where the estimated best action is selected. They show prove a regret bound on the order of $O((\|\theta_\star\| + 1/\|\theta_\star\|)d\sqrt{n})$ for their approach, which notably behaves poorly as $\theta \to 0$. This behaviour is because their exploration is isotropic—equal in all directions—and not directed by an estimate of $\theta_\star$. In contrast, randomised exploration algorithms account for structure by (i) taking $X_t = \nabla J(\theta_t)$ (almost surely), which accounts for the geometry of the action-set, and (ii) sampling $\theta_t$ from a distribution concentrated on a scaled version of $\Theta_{t-1}$, which accounts for the current estimate of $\theta_\star$. One might interpret randomised algorithms as blending together the exploration and exploitation stages with a more careful balance between the two.

## 5. Proof of main result

We now prove our main result, Theorem 4. Here and in the appendices, we will write $J$ in place of $J_\mathcal{X}$, and we will work throughout on the $1 - \delta$ probability event where $\theta_\star \in \cap_{t \geq 1}\Theta_{t-1}$.

We start by moving from $R_n$ to $\bar{R}_n := \sum_{t=1}^n \mathbb{E}_{t-1} r_t$. This can be done by noting that $\xi_t = r_t - \mathbb{E}_{t-1} r_t$, $t \geq 1$, is a martingale difference sequence satisfying $|\xi_t| \lesssim \sqrt{d}$ for all $t \geq 1$, and applying a standard concentration inequality (included here as Lemma 8, Appendix A). From this, conclude that with probability $1 - \delta$, for all $n \geq 1$,

$$R_n \lesssim \bar{R}_n + \sqrt{dn \log(dn/\delta)}\,.$$

We now outline the three main results we use in bounding $\bar{R}_n$, and then show how they come together.

### 5.1. Regret decomposition & upper bound

Denote by $p_{t-1}$ the conditional probability of optimism $\mathbb{P}_{t-1}\{J(\theta_t) \geq J(\theta_\star)\}$ at time-step $t \geq 1$. Letting $\chi_{t-1} = \mathbf{1}[p_{t-1} \leq p]$ for a threshold $p \in (0, 1)$, we now decompose the regret into that incurred in time-steps where $p_{t-1}$ is high, and those where it is low (we take $p = 1/(16K^4)$, where $K$ is the constant appearing in Assumption 1):

$$
\begin{aligned}
\bar{R}_n &= \sum_{t=1}^n \chi_{t-1}\mathbb{E}_{t-1} r_t + \sum_{t=1}^n (1 - \chi_{t-1})\mathbb{E}_{t-1} r_t \\
&\lesssim \underbrace{\frac{M(\beta_n^2 \vee K^2 d)}{J(\theta_\star)} \sum_{t=1}^n \chi_{t-1} \sup_{u \in \mathbb{B}_2^d} \|V_{t-1}^{-1/2} u\|_*^2}_{=:\bar{R}_n^{\mathrm{TS}}} + \underbrace{K^4(\beta_n \vee K\sqrt{d}) \sum_{t=1}^n \mathbb{E}_{t-1}\|X_t\|_{V_{t-1}^{-1}}}_{=:\bar{R}_n^{\mathrm{OPT}}}\,. \quad (3)
\end{aligned}
$$

The derivations of the bound is presented in Appendix B. It is based on repeatedly applying properties of Bregman divergences and convex functions. At a high level, we introduce $\theta_t'$, which is, conditionally on $\mathbb{F}_{t-1}$, an independent copy of $\theta_t$; then conditioning condition on the event $\{J(\theta_t') \leq J(\theta_\star)\}$ (the converse for the second term), and integrating the $\theta_t'$ out.

Examining the two terms, $\bar{R}_n^{\text{OPT}}$ is a term that appears in the standard regret analysis of optimistic algorithms, and is easily handled using a concentration argument (Lemma 9) and the elliptical potential lemma (Lemma 10); this yields

$$\bar{R}_n^{\text{OPT}} \lesssim (\beta_n \vee K\sqrt{d})K^4\sqrt{n(d\log(1 + n/(d\lambda)) + \log(1/\delta))}\,,$$

a term featuring in our overall regret bound. The term $\bar{R}_n^{\text{TS}}$ is a cost associated with randomised exploration: it is the sum of the sizes of the parameter sampling distributions (or confidence sets, as these are the same up to scaling), where size is measured in the geometry induced by $\|\cdot\|_*$.

## 5.2. Relating confidence widths to the amount of exploration

The challenge is now to show that $V_t$ grows sufficiently fast, measured with respect to the geometry induced by $\|\cdot\|_*$, such that $\bar{R}_n^{\text{TS}}$ is small. First, we relate the width $\|V_{t-1}^{-1/2}u\|_*$ to the expected amount of exploration in the direction of $u \in \mathbb{B}_2^d$ at step $t$, with the latter measured in the $\ell_2$ norm, $\|\cdot\|$, at a cost of $1/m$ from $m$-strong convexity. This is a *change of geometry* lemma:

**Lemma 5.** *For all $t \geq 1$ with $p_{t-1} \leq 1/(16K^4)$, for any $u \in \mathbb{B}_2^d$,*

$$\frac{1}{J(\theta_\star)}\|V_{t-1}^{-1/2}u\|_*^2 \lesssim \frac{K}{m}\|\mathbb{E}_{t-1}[X_tX_t^\mathsf{T}]^{1/2}V_{t-1}^{-1/2}u\|\,.$$

**Remark 3.** *When $\mathcal{X} = \mathbb{B}_2^d$, we have $m = 1$ for $\|\cdot\|_* = \|\cdot\|$, and thus no change of geometry is needed. In that case $X_t = \theta_t/\|\theta_t\|$ almost surely, and $J(\theta) = \|\theta\|$ for all $\theta \in \mathbb{R}^d$, and so*

$$\mathbb{E}_{t-1}X_tX_t^\mathsf{T} = \mathbb{E}_{t-1}[\theta_t\theta_t^\mathsf{T}/\|\theta_t\|^2] \approx \frac{1}{\|\theta_\star\|^2}\mathbb{E}_{t-1}\theta_t\theta_t^\mathsf{T} \succeq \frac{1}{\|\theta_\star\|^2}\text{Var}_{t-1}\,\theta_t = \frac{K^2}{J^2(\theta_\star)}V_{t-1}^{-1}\,,$$

*where we for the sake of exposition, allowed ourselves the simplifying assumption our the confidence sets and perturbations are concentrated sufficiently to ensure that $1/\|\theta_t\| \approx 1/\|\theta_\star\|$.*

We present the proof of Lemma 5 in Appendix C. Once again, we proceed by introducing a random variable $\theta_t'$ with the same $\mathbb{F}_{t-1}$-conditional law as $\theta_t$; however, this time, we couple $\theta_t$ and $\theta_t'$ closely coupled, in that they differ only in the $u$ marginal (along which they are independent). We then proceed with a convex Poincaré inequality-style argument along the $u$ direction, which relates $X_t = \nabla J(\theta_t)$ and the $V_{t-1}^{-1}$ matrix, with the latter being essentially the conditional variance of $\theta_t$.

## 5.3. Establishing the growth of the design matrices

The final ingredient is the following relation between the sum $\sum_{t=1}^n \mathbb{E}_{t-1}X_tX_t^\mathsf{T}$ of the conditional expected increments in the design matrices and their realisation $\sum_{t=1}^n X_tX_t$.

**Lemma 6.** *For any $r \in (0, 1]$ and $\delta \in (0, 1)$, with probability at least $1 - \delta$, for all $n \geq 1$ and all $u \in r\mathbb{B}_2^d$,*

$$u^\mathsf{T}\sum_{i=1}^n X_tX_t^\mathsf{T}u + J^2(u)\omega_n + 5 \geq \frac{1}{2}\sum_{t=1}^n u^\mathsf{T}\mathbb{E}_{t-1}[X_tX_t^\mathsf{T}]u\,,$$

*where $\omega_n = d\log(20d^3n^2r/\delta^2)$.*

**Remark 4.** *A standard matrix Chernoff inequality[1] gives that with probability $1 - \delta$, for all $n \geq 1$,*

$$\sum_{t=1}^{n} X_t X_t + \log(d/\delta) \succeq \frac{1}{2} \sum_{t=1}^{n} \mathbb{E}_{t-1} X_t X_t^{\mathsf{T}}, \tag{4}$$

*where $\succeq$ denotes the usual ordering on positive-semidefinite matrices. For the $\ell_2$ ball, Equation (4) serves the same role as Lemma 6, but is tighter. However, in the general setting where $J(u) \neq \|u\|$, it is crucial that we obtain the $J^2(u)$ dependence seen in Lemma 6.*

The proof of Lemma 6 is presented in Appendix D. It uses Lemma 9, a one-dimensional version of the inequality given in Equation (4), and applies it to the process $(\langle u, X_t \rangle^2 \colon t \geq 1)$ for all $u$ in a time-dependent cover of $r\mathbb{B}_2^d$. A union bound over the size of the cover is responsible for the $\omega_n$ term, and the discretisation error involved in the covering argument yields the additive 5.

### 5.4. Putting everything together

Let $N_n = \sum_{t=1}^{n} \chi_{t-1}$ be the number of steps up to $n$ on which the conditional probability of optimism was below the threshold $p$. We will shortly show that Lemma 5 and Lemma 6, together with the assumed smoothness, yield the following bound:

**Claim 7.** *For all $t \geq 2$ and $u \in \mathbb{B}_2^d$,*

$$\frac{1}{J(\theta_\star)} \|V_{t-1}^{-1/2} u\|_*^2 \lesssim \frac{M \omega_{t-1} K^2 J(\theta_\star)}{m^2 N_{t-1}} + \frac{K}{m \sqrt{N_{t-1}}}.$$

First though, note that using Claim 7 within the regret decomposition of Equation (3) completes the proof. Indeed, using that the expected per-step regret is bounded by $2\|\theta_\star\| \lesssim \sqrt{d}$ (to handle step the first step, which is not covered by Claim 7), and the usual integral for monotonic integrands, we have

$$\bar{R}_n^{\mathrm{TS}} \lesssim \sqrt{d} + (\beta_n^2 \vee K^2 d) \int_1^n \left[ \frac{M^2 \omega_t K^2 J(\theta_\star)}{m^2 N_t} + \frac{K}{m \sqrt{N_t}} \right] dt$$

$$\lesssim \sqrt{d} + d\sqrt{d}(\beta_n^2 \vee K^2 d) K^2 \frac{M^2}{m^2} \log(dn/(\delta\lambda)) \log n + \frac{M}{m} K(\beta_n^2 \vee K^2 d) \sqrt{n},$$

which completes our bound (observe that the first two terms are lower order).

*Proof of Claim 7.* We work on the $1 - \delta$ probability event resulting from applying Lemma 6 with $r = 1/\sqrt{\lambda}$. Since $u \in \mathbb{B}_2^d$, we have $V_n^{-1/2} u \in \mathbb{B}_2^d/\sqrt{\lambda}$, and thus for all $n \geq 1$,

$$J^2(V_n^{-1/2} u)\omega_n + 6 \geq \frac{1}{2} \sum_{t=1}^{n} \|\mathbb{E}_{t-1}[X_t X_t^{\mathsf{T}}]^{1/2} V_n^{-1/2} u\|^2. \tag{5}$$

Now we proceed to in turn upper and lower-bounding the above expression.

For the upper-bound, note that by $M$-smoothness, $J^2(V_n^{-1/2} u) \leq \frac{M}{2} \|V_n^{-1/2} u\|_*^2$.

---

1. See Tropp (2012). This exact inequality is not stated there, but all the tools needed to derive it are.

For the lower bound of the right-hand side of Equation (5), we will use Lemma 5. Let $v_{t-1} = V_{t-1}^{1/2} V_n^{-1/2} u$, and note that since $V_{t-1} \preceq V_n$, we have that $\|v_{t-1}\| \leq 1$. Now,

$$
\begin{aligned}
\sum_{t=1}^n \chi_{t-1} \| \mathbb{E}_{t-1}[X_t X_t^{\mathsf{T}}]^{1/2} V_n^{-1/2} u \|^2 &= \sum_{t=1}^n \chi_{t-1} \|v_{t-1}\|^2 \| \mathbb{E}_{t-1}[X_t X_t^{\mathsf{T}}]^{1/2} V_{t-1}^{-1/2} \frac{v_{t-1}}{\|v_{t-1}\|} \|^2 \\
&\gtrsim \frac{m^2}{K^2 J^2(\theta_\star)} \sum_{t=1}^n \chi_{t-1} \frac{\|V_{t-1}^{-1/2} v_{t-1}\|_*^4}{\|v_{t-1}\|^2} \qquad \text{(Lemma 5)} \\
&\geq \frac{m^2}{K^2 J^2(\theta_\star)} N_n \|V_n^{-1/2} u\|_*^4 \qquad\qquad (\|v_{t-1}\| \leq 1)
\end{aligned}
$$

Combining our lower and upper bounds on Equation (5), writing $\alpha_n = Cm^2 N_n / K^2$ for a numerical constant $C > 0$ and letting $y = \frac{1}{J(\theta_\star)} \|V_n^{-1/2} u\|_*^2$, we obtain the quadratic

$$
-\alpha_n y^2 + M\omega_n J(\theta_\star) y + 6 \geq 0 \,.
$$

Solving for $y$, we have

$$
\|V_n^{-1/2} u\|_*^2 \leq \frac{M\omega_n J(\theta_\star) + \sqrt{M^2 \omega_n^2 J^2(\theta_\star) + 24\alpha_n}}{2\alpha_n} \leq \frac{M\omega_n J(\theta_\star)}{\alpha_n} + \sqrt{\frac{6}{\alpha_n}} \,,
$$

whence relabelling $n \mapsto t - 1$ concludes the proof. $\qquad\qquad\square$

## 6. Conclusion

In this paper, we have presented a new analysis of randomised exploration algorithms for the linear bandit setting, which establishes that, given a nice-enough action set, randomised algorithms can obtain the optimal dependence on the dimension of the problem without need for any algorithmic modifications. Our improved regret bounds requires that the action space satisfies a smoothness and strong convexity condition, Assumption 2, which ensures that small perturbations in the parameter space are translate directly to at least some perturbations in the action space, while also guaranteeing that these do not lead to large changes in the instantaneous regret.

Our results complement the lower bounds by Hamidi and Bayati (2023) and Zhang (2022) which show that linear Thompson sampling can suffer linear regret in particular settings where the connection between randomness in the parameter and action spaces is broken. However, these results together still do not give a complete characterisation of when randomised exploration algorithms can and cannot achieve the optimal rate of regret in the linear bandit setting: it remains an important open problem to understand exactly which action spaces permit an optimal dependence on the dimension.

## Acknowledgements

This project started in earnest as a result of discussions taking place at the 2023 Workshop on the Theory of Reinforcement Learning in Edmonton. We thank Csaba Szepesvári for organising this workshop, and for feedback on early versions of this work. DJ & MA thank Gergely Neu for putting them in touch with CP-B, who was working contemporaneously on the same problem.

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

## Appendix A. Some standard results

The following lemma is adapted from Exercise 20.8 in Lattimore and Szepesvári (2020).

**Lemma 8.** *Fix $0 < \delta \leq 1$. Let $(\xi_t)_{t \in \mathbb{N}^+}$ be a real-valued martingale difference sequence satisfying $|\xi_t| \leq c$ almost surely for each $t \in \mathbb{N}^+$ and some $c > 0$. Then,*

$$\mathbb{P}\left(\exists n\colon \left(\sum_{t=1}^{n} \xi_t\right)^2 \geq 2(c^2 n + 1) \log\left(\sqrt{c^2 n + 1}/\delta\right)\right) \leq \delta.$$

Next is a second concentration inequality that we require.

**Lemma 9.** *Let $(\alpha_t)_{t \geq 1}$ be a sequence of random variables adapted to a filtration $(\mathbb{F}_t)_{t \geq 0}$ with $0 \leq \alpha_t \leq R$ for all $t \geq 0$. Then, for all $\delta \in (0, 1)$,*

$$\mathbb{P}\left\{\sum_{t=1}^{n} \alpha_t \geq (1 - 1/e) \sum_{t=1}^{n} \mathbb{E}[\alpha_t \mid \mathbb{F}_{t-1}] - R \log 1/\delta, \ \forall n \geq 1\right\} \geq 1 - \delta.$$

*Proof of Lemma 9.* By rescaling, we need only consider the case where $R = 1$. Let $(S_n)_{n \geq 0}$ be the random process defined by $S_0 = 0$ and

$$S_n = \exp\left((1 - 1/e) \sum_{t=1}^{n} \mathbb{E}[\alpha_t \mid \mathbb{F}_{t-1}] - \sum_{t=1}^{n} \alpha_t\right).$$

Observe that $S_n \geq 0$ for all $n \geq 0$, and that since for any $n \geq 0$,

$$\mathbb{E}[\exp\{-\alpha_{n+1}\} \mid \mathbb{F}_n] \leq 1 - (1 - 1/e)\mathbb{E}[\alpha_{n+1} \mid \mathbb{F}_n] \leq \exp\{-(1 - 1/e)\mathbb{E}[\alpha_{n+1} \mid \mathbb{F}_n]\}$$

we have that for al $n \geq 0$,

$$\mathbb{E}[S_{n+1} \mid \mathbb{F}_n] = S_n \exp\{(1 - 1/e)\mathbb{E}[\alpha_{n+1} \mid \mathbb{F}_n] - \alpha_{n+1}\} \leq S_n.$$

Therefore, $(S_n)_{n \geq 0}$ is a non-negative supermartingale. Applying Ville's inequality yields the result. □

The following is an adaptation of Lemma 19.4 in Lattimore and Szepesvári (2020),

**Lemma 10** (Elliptical potential lemma). *Fix $\lambda > 0$ and a sequence $a_1, a_2, \ldots$ in $\mathbb{B}_2^d$. Then, letting $V_n = \sum_{t=1}^{n} a_t a_t^\mathsf{T} + \lambda I$, we have that for all $n \geq 1$,*

$$\sum_{t=1}^{n} \|a_t\|_{V_{t-1}^{-1}}^2 \leq 2d \log(1 + n/(d\lambda)).$$

## Appendix B. Derivation of the regret decomposition upper bound (Equation (3))

Let $p_{t-1} = \mathbb{P}_{t-1}\{J(\theta_t) \geq J(\theta_\star)\}$ be the (conditional) probability of optimism at step $t \geq 1$, and let $\tilde{\beta}_{t-1} = \beta_{t-1} \vee K\sqrt{d}$. We now derive two bounds separately. When $p_{t-1}$ is high, we will use

$$\mathbb{E}_{t-1} D_J(\theta_\star, \theta_t) \leq \frac{4\tilde{\beta}_{t-1}}{p_{t-1}} \mathbb{E}_{t-1}\left[\|X_t\|^2_{V_{t-1}^{-1}}\right]^{\frac{1}{2}}. \tag{6}$$

When $p_{t-1}$ is low, we will prefer the bound

$$\mathbb{E}_{t-1} D_J(\theta_\star, \theta_t) \leq \frac{1}{1-p_{t-1}}\left\{\frac{4\pi\tilde{\beta}_{t-1}^2}{J(\theta_\star)}\sup_{u\in\mathbb{B}_2^d}\|V_{t-1}^{-1/2}u\|_*^2 + 6\tilde{\beta}_{t-1}\mathbb{E}_{t-1}\left[\|X_t\|^2_{V_{t-1}^{-1}}\right]^{\frac{1}{2}}\right\}. \tag{7}$$

Combining these two bounds with our regret decomposition establishes Equation (3)

We will derive the two bounds Equations (6) and (7) using similar techniques. Let $P_{t-1}(A) = \mathbb{P}_{t-1}\{\theta_t \in A\}$. Both derivations will make use of the following estimates.

**Claim 11.** *For any norm $F$ on $\mathbb{R}^d$,*

$$\int F^2(\theta_\star - a)P_{t-1}(da) \vee \int F^2(a-b)P_{t-1}^2(da \times db) \leq 4(\beta_{t-1}^2 \vee K^2 d)\sup_{u\in\mathbb{B}_2^d} F^2(V_{t-1}^{-1/2}u).$$

*Proof.* Letting $\eta_a$ and $\eta_b$ be independent copies of $\eta_t$, we can express $a$ and $b$ as

$$a = \hat{\theta}_{t-1} + V_{t-1}^{-1/2}\eta_a \quad \text{and} \quad b = \hat{\theta}_{t-1} + V_{t-1}^{1/2}\eta_b.$$

Denote by $\mathbb{E}_{\eta_a,\eta_b}$ the expectation over $\eta_a$ and $\eta_b$. We have that

$$\int F^2(a-b)P_{t-1}^2(da \times db) = \mathbb{E}_{\eta_a,\eta_b} F^2(V_{t-1}^{-1/2}(\eta_a - \eta_b))$$

$$\leq 4\mathbb{E}_{\eta_a}\|\eta_a\|^2 \sup_{u\in\mathbb{B}_2^d} F^2(V_{t-1}^{-1/2}u)$$

$$\leq 4K^2 d \sup_{u\in\mathbb{B}_2^d} F^2(V_{t-1}^{-1/2}u),$$

where we used that $\mathbb{E}_{\eta_b}\|\eta_b\| = \mathbb{E}_{\eta_a}\|\eta_a\|^2 = \mathbb{E}_{t-1}\|\eta_t\|^2 \leq K^2 d$.

Expressing $\theta_\star = \hat{\theta}_{t-1} + \beta_{t-1}V_{t-1}^{-1/2}u'$ for some $u' \in \mathbb{B}_2^d$ (which we can do due to the implicit assumption that $\theta_\star \in \Theta_{t-1}$) and using the same approach we obtain the other part of the bound. $\square$

*Derivation of Equation (6).* For almost every $\theta_t, \theta_t' \in \mathbb{R}^d$ such that $J(\theta_t') \geq J(\theta_\star)$,

$$D_J(\theta_\star, \theta_t) \leq J(\theta_t') - J(\theta_t) - \langle\nabla J(\theta_t), \theta_\star - \theta_t\rangle \qquad (J(\theta_t') \geq J(\theta_\star))$$

$$\leq \langle\nabla J(\theta_t'), \theta_t' - \theta_t\rangle - \langle\nabla J(\theta_t), \theta_\star - \theta_t\rangle \qquad \text{(convexity)}$$

$$\leq \|\nabla J(\theta_t')\|_{V_{t-1}^{-1}}\|\theta_t' - \theta_t\|_{V_{t-1}} + \|\nabla J(\theta_t)\|_{V_{t-1}^{-1}}\|\theta_\star - \theta_t\|_{V_{t-1}}. \quad \text{(Cauchy-Schwarz)}$$

Now let $Q$ be a measure on $\mathbb{R}^d$ given by

$$Q(A) = \begin{cases} \frac{1}{p_{t-1}}P_{t-1}(A \cap \{\theta \in \mathbb{R}^d \colon J(\theta) \geq J(\theta_\star)\}), & p_{t-1} \neq 0 \\ \text{any arbitrary measure} & \text{otherwise.} \end{cases}$$

Since the bound above holds for almost all $\theta_t' \in \mathbb{R}^d$ such that $J(\theta_t') \geq J(\theta_\star)$, and $Q$ is a diffuse measure on that set, it also holds on average for $\theta_t' \sim Q$. Integrating with respect to $Q$ and $P_{t-1}$,

$$\mathbb{E}_{t-1} D_J(\theta_\star, \theta_t) \leq \int \|\nabla J(\theta_t')\|_{V_{t-1}^{-1}} \|\theta_t' - \theta_t\|_{V_{t-1}} (P_{t-1} \otimes Q)(d\theta_t \times d\theta_t')$$
$$+ \int \|\nabla J(\theta_t)\|_{V_{t-1}^{-1}} \|\theta_\star - \theta_t\|_{V_{t-1}} P_{t-1}(d\theta_t) .$$

For the first integral,

$$\int \|\nabla J(\theta_t')\|_{V_{t-1}^{-1}} \|\theta_t' - \theta_t\|_{V_{t-1}} (P_{t-1} \otimes Q)(d\theta_t \times d\theta_t') \tag{8}$$
$$\leq \frac{1}{p_{t-1}} \int \|\nabla J(\theta_t')\|_{V_{t-1}^{-1}} \|\theta_t' - \theta_t\|_{V_{t-1}} P_{t-1}^2(d\theta_t \times d\theta_t') \quad (\forall f \geq 0, \int f dQ \leq \frac{1}{p_{t-1}} \int f dP_{t-1})$$
$$\leq \frac{1}{p_{t-1}} \left[ \int \|\nabla J(\theta_t')\|_{V_{t-1}^{-1}}^2 P_{t-1}(\theta_t') \int \|\theta_t' - \theta_t\|_{V_{t-1}}^2 P_{t-1}^2(d\theta_t \times d\theta_t') \right]^{1/2} \quad \text{(Cauchy-Schwarz)}$$
$$\leq \frac{2(\beta_{t-1} \vee K\sqrt{d})}{p_{t-1}} \left[ \int \|\nabla J(\theta_t)\|_{V_{t-1}^{-1}}^2 P_{t-1}(\theta_t) \right]^{1/2} . \quad \text{(Claim 11)}$$

Finally, since $\nabla J(\theta_t) = X_t$ almost surely, $\mathbb{E}_{t-1}[\|\nabla J(\theta_t)\|_{V_{t-1}^{-1}}^2]^{1/2} = \mathbb{E}_{t-1}[\|X_t\|_{V_{t-1}^{-1}}^2]^{1/2}$.

The second integral follows likewise, with the addition of multiplying the resulting nonnegative bound by $1/p_{t-1} \geq 1$ to keep things tidy. $\qquad \square$

For the steps with a low probability of optimism, we will need the following property of Bregman divergences:

**Lemma 12** (Law of cosines). *For any convex function $f \colon \mathbb{R}^d \to \mathbb{R}$ and all $x$ and almost all $y, z \in \mathbb{R}^d$,*

$$D_f(x, y) = D_f(x, z) + D_f(z, y) - \langle x - z, \nabla f(y) - \nabla f(z) \rangle .$$

*Derivation of Equation (7).* For almost all $\theta_t, \theta_t' \in \mathbb{R}^d$,

$$\begin{aligned} D_J(\theta_\star, \theta_t) &= D_J(\theta_\star, \theta_t') + D_J(\theta_t', \theta_t) - \langle \theta_\star - \theta_t', \nabla J(\theta_t) - \nabla J(\theta_t') \rangle && \text{(law of cosines)} \\ &\leq D_J(\theta_\star, \theta_t') + \langle \theta_\star - \theta_t, \nabla J(\theta_t') - \nabla J(\theta_t) \rangle && \text{(convexity of } J \text{ in } D_J) \\ &\leq D_J(\theta_\star, \theta_t') + \|\nabla J(\theta_t') - \nabla J(\theta_t)\|_{V_{t-1}^{-1}} \|\theta_\star - \theta_t\|_{V_{t-1}} . && \text{(Cauchy-Schwartz)} \end{aligned}$$

Also, for almost every $\theta_t' \in \mathbb{R}^d$ satisfying $J(\theta_t') \leq J(\theta_\star)$,

$$
\begin{aligned}
D_J(\theta_\star, \theta_t') &= J(\theta_\star) - J(\theta_t') - \langle \nabla J(\theta_t'), \theta_\star - \theta_t' \rangle \\
&= \frac{1}{J(\theta_\star)} \left[ J^2(\theta_\star) - J(\theta_t')J(\theta_\star) - \langle 2J(\theta_t')\nabla J(\theta_t'), \theta_\star - \theta_t' \rangle \right] \\
&\quad + \left( 2\frac{J(\theta_t')}{J(\theta_\star)} - 1 \right) \langle \nabla J(\theta_t'), \theta_\star - \theta_t' \rangle \\
&\leq \frac{1}{J(\theta_\star)} \left[ J^2(\theta_\star) - J^2(\theta_t') - \langle 2J(\theta_t')\nabla J(\theta_t'), \theta_\star - \theta_t' \rangle \right] + |\langle \nabla J(\theta_t'), \theta_\star - \theta_t' \rangle| \\
&\qquad\qquad\qquad\qquad\qquad\qquad\qquad\qquad\qquad\qquad (0 < J(\theta_t') \leq J(\theta_\star)) \\
&= \frac{1}{J(\theta_\star)} D_{J^2}(\theta_\star, \theta_t') + |\langle \nabla J(\theta_t'), \theta_\star - \theta_t' \rangle| \qquad (2J(\theta_t')\nabla J(\theta_t') = \nabla J^2(\theta_t') \text{ a.e.}) \\
&\leq \frac{1}{J(\theta_\star)} D_{J^2}(\theta_\star, \theta_t') + \|\nabla J(\theta_t')\|_{V_{t-1}^{-1}} \|\theta_\star - \theta_t'\|_{V_{t-1}} . \qquad \text{(Cauchy-Schwartz)}
\end{aligned}
$$

Combining the above two bounds, we have that for almost all $\theta_t, \theta_t' \in \mathbb{R}^d$, if $J(\theta_t') \leq J(\theta_\star)$, then

$$
\begin{aligned}
D_J(\theta_\star, \theta_t) &\leq \frac{1}{J(\theta_\star)} D_{J^2}(\theta_\star, \theta_t') + \|\nabla J(\theta_t)\|_{V_{t-1}^{-1}} \|\theta_\star - \theta_t\|_{V_{t-1}} \\
&\quad + \|\nabla J(\theta_t')\|_{V_{t-1}^{-1}} \left[ \|\theta_\star - \theta_t\|_{V_{t-1}} + \|\theta_\star - \theta_t'\|_{V_{t-1}} \right] . \qquad (9)
\end{aligned}
$$

Now let $Q$ be a measure on $\mathbb{R}^d$ given by

$$
Q(A) = \begin{cases} \frac{1}{1-p_{t-1}} P_{t-1}(A \cap \{\theta \in \mathbb{R}^d \colon J(\theta) \leq J(\theta_\star)\}), & p_{t-1} \neq 1 \\ \text{any arbitrary measure} & \text{otherwise.} \end{cases}
$$

Since Equation (9) holds for almost all $\theta_t, \theta_t' \in \mathbb{R}^d$ with $J(\theta_t') \leq J(\theta_\star)$ and $Q, P_{t-1}$ are non-atomic, it also holds on average for $\theta_t' \sim Q$ and $\theta_t \sim P_{t-1}$. Integrating, we see that $\mathbb{E}_{t-1} D_J(\theta_\star, \theta_t)$ is upper bounded by

$$
\begin{aligned}
&\frac{1}{J(\theta_\star)} \int D_{J^2}(\theta_\star, \theta_t') Q(d\theta_t') + \int \|\nabla J(\theta_t')\|_{V_{t-1}^{-1}} Q(d\theta_t') \int \|\theta_\star - \theta_t\|_{V_{t-1}} P_{t-1}(d\theta_t) \qquad (10) \\
&+ \int \|\nabla J(\theta_t')\|_{V_{t-1}^{-1}} \|\theta_\star - \theta_t'\|_{V_{t-1}} Q(d\theta_t') + \int \|\nabla J(\theta_t)\|_{V_{t-1}^{-1}} \|\theta_\star - \theta_t\|_{V_{t-1}} P_{t-1}(d\theta_t) .
\end{aligned}
$$

For the first integral, we can use that for any $f \geq 0$, $\int f dQ \leq \frac{1}{1-p_{t-1}} \int f dP_{t-1}$, to establish that

$$
\int D_{J^2}(\theta_\star, \theta_t') Q(d\theta_t') \leq \frac{1}{(1-p_{t-1})} \int D_{J^2}(\theta_\star, \theta_t') P_{t-1}(d\theta_t') = \frac{1}{(1-p_{t-1})} \mathbb{E}_{t-1} D_{J^2}(\theta_\star, \theta_t) ,
$$

where the final equality follows since $\theta_t' \sim P_{t-1}$ has the same law as $\theta_t$ conditioned on $\mathbb{F}_{t-1}$. Now, by Assumption 2 and then using the estimate from Claim 11,

$$
\mathbb{E}_{t-1} D_{J^2}(\theta_\star, \theta_t) \leq \pi \mathbb{E}_{t-1} \|\theta_\star - \theta_t\|_\star^2 \leq 4\pi(\beta_{t-1}^2 \vee K^2 d) \sup_{u \in \mathbb{B}_2^d} \|V_{t-1}^{-1/2} u\|_\star^2 ,
$$

Bounding the remaining integrals in Equation (10) can be done by following the same steps as for the integral in Equation (8) of the optimistic bound, just with $\frac{1}{1-p_{t-1}}$ in place of $\frac{1}{p_{t-1}}$. $\qquad \square$

## Appendix C. Proof of the change of geometry lemma (Lemma 5)

Let $u^\perp \colon \mathbb{R}^d \to \mathbb{R}^d$ be a basis completion orthogonal to $u$ (a projection onto the orthogonal complement of the span of $u$). Let $\epsilon = \langle \eta_t, u \rangle / \|u\|$, and let $\tilde{\epsilon}$ be an independent copy of $\epsilon$ independent of $\mathbb{F}_{t-1}$ and define

$$\tilde{\theta}_t = \hat{\theta}_{t-1} + V_{t-1}^{-1/2} u^\perp \eta_t + V_{t-1}^{-1/2} u \tilde{\epsilon} \quad \text{observing that} \quad \theta_t - \tilde{\theta}_t = (\epsilon - \tilde{\epsilon}) u \,.$$

Also define the indicators $\iota = \mathbf{1}[J(\theta_t) \le J(\theta_\star)]$ and $\tilde{\iota} = \mathbf{1}[J(\tilde{\theta}_t) \le J(\theta_\star)]$.

*Proof of Lemma 5.* The proof is based on lower and upper-bounding $\mathbb{E}_{t-1} \iota \tilde{\iota} D_{J^2}(\tilde{\theta}_t, \theta_t)$.

For the lower bound, note that by strong convexity,

$$\mathbb{E}_{t-1} \iota \tilde{\iota} D_{J^2}(\tilde{\theta}_t, \theta_t) \ge \frac{m}{2} \|V_{t-1}^{-1/2} u\|_*^2 \, \mathbb{E}_{t-1} \iota \tilde{\iota} (\tilde{\epsilon} - \epsilon)^2 \,,$$

where

$$
\begin{aligned}
\mathbb{E}_{t-1} \iota \tilde{\iota} (\tilde{\epsilon} - \epsilon)^2 &= \mathbb{E}_{t-1} (\tilde{\epsilon} - \epsilon)^2 - \mathbb{E}_{t-1}((\iota + \tilde{\iota}) \wedge 1)(\tilde{\epsilon} - \epsilon)^2 \\
&\ge 2 - \mathbb{E}_{t-1}((\iota + \tilde{\iota}) \wedge 1)(\tilde{\epsilon} - \epsilon)^2 && \text{(marginal variance assumption)} \\
&\ge 2 - 2\mathbb{E}_{t-1} \iota (\tilde{\epsilon} - \epsilon)^2 && \text{(drop $\wedge$)} \\
&\ge 2 - 2\mathbb{E}_{t-1} \iota (K^2 + \epsilon^2) && \text{(marginal variance assumption)} \\
&\ge 2 - 2\sqrt{p_{t-1}} \sqrt{\mathbb{E}_{t-1}(K^2 + \epsilon^2)^2} && \text{(Cauchy-Schwarz and $\mathbb{E}_{t-1} \iota = p_{t-1}$)} \\
&\ge 2 - 4K^2 \sqrt{p_{t-1}} && \text{(marginal variance and fourth moment assumptions)} \\
&\ge 1 && \text{($p_{t-1} \le p = 1/(16K^4)$ by assumption)}
\end{aligned}
$$

For the upper bound, we have that

$$
\begin{aligned}
\mathbb{E}_{t-1} \iota \tilde{\iota} D_{J^2}(\tilde{\theta}_t, \theta_t) &= \mathbb{E}_{t-1} \iota \tilde{\iota} (J^2(\tilde{\theta}_t) - J^2(\theta_t) - \langle \nabla J^2(\theta_t), \tilde{\theta}_t - \theta_t \rangle) \\
&= \mathbb{E}_{t-1} \iota \tilde{\iota} (J^2(\tilde{\theta}_t) - J^2(\theta_t) - \langle 2J(\theta_t) \nabla J(\theta_t), \tilde{\theta}_t - \theta_t \rangle) \\
&\le \mathbb{E}_{t-1} \iota \tilde{\iota} |J^2(\tilde{\theta}_t) - J^2(\theta_t)| + 2J(\theta_\star) \mathbb{E}_{t-1} |\langle \nabla J(\theta_t), \tilde{\theta}_t - \theta_t \rangle| \\
&&\hspace{-3cm} (0 < \iota J(\theta_t) \le J(\theta_\star)) \\
&= \mathbb{E}_{t-1} \iota \tilde{\iota} |J(\tilde{\theta}_t) - J(\theta_t)|(J(\theta_t) + J(\tilde{\theta}_t)) + 2J(\theta_\star) \mathbb{E}_{t-1} |\langle \nabla J(\theta_t), \tilde{\theta}_t - \theta_t \rangle| \\
&\le 2J(\theta_\star) \left\{ \mathbb{E}_{t-1} |J(\tilde{\theta}_t) - J(\theta_t)| + \mathbb{E}_{t-1} |\langle \nabla J(\theta_t), \tilde{\theta}_t - \theta_t \rangle| \right\} \\
&\le 6J(\theta_\star) \mathbb{E}_{t-1} |\langle \nabla J(\theta_t), \tilde{\theta}_t - \theta_t \rangle| && \text{(convexity)} \\
&= 6J(\theta_\star) \mathbb{E}_{t-1} (\tilde{\epsilon} - \epsilon) |\langle \nabla J(\theta_t), V_{t-1}^{-1/2} u \rangle| \\
&\le 6J(\theta_\star) \mathbb{E}_{t-1}[(\tilde{\epsilon} - \epsilon)^2]^{1/2} \|\mathbb{E}_{t-1}[\nabla J(\theta_t) \nabla J(\theta_t)^\mathsf{T}]^{1/2} V_{t-1}^{-1/2} u\| \\
&&\hspace{-3cm} \text{(Cauchy-Schwarz)} \\
&\le 6\sqrt{2} J(\theta_\star) K \|\mathbb{E}_{t-1}[\nabla J(\theta_t) \nabla J(\theta_t)^\mathsf{T}]^{1/2} V_{t-1}^{1/2} u\| \,. \\
&&\hspace{-3cm} \text{(marginal variance assumption)}
\end{aligned}
$$

Chaining the lower and upper bounds yields the claimed result. $\qquad\square$

## Appendix D.  Proof of directional concentration (Lemma 6)

**Lemma 13.** *For any $r, \epsilon > 0$, the covering number of $r\mathbb{B}_2^d$ is upper bounded by $r^d(1 + \frac{2}{\epsilon})^d$.*

*Proof of Lemma 6.* For each $n \geq 1$, let $\mathcal{N}_n$ be a minimal $\epsilon_n$-cover of $r\mathbb{B}_2^d$ in $\|\cdot\|$, where the value of $\epsilon_n > 0$ will be chosen shortly. Let

$$\Delta_n = \sum_{t=1}^n X_t X_t^\mathsf{T} - (1 - 1/e) \sum_{t=1}^n \mathbb{E}_{t-1}[X_t X_t^\mathsf{T}].$$

For every $n \geq 1$ and $u \in \mathcal{N}_n$, we apply Lemma 9 to the sequence $\alpha_t = \langle X_t, u \rangle^2$, $t \geq 1$, using the upper bound $\alpha_t \leq J(u)^2$ for all $t \geq 1$, and confidence level $\delta_n = 6\delta/(\pi^2 n^2 |\mathcal{N}_n|)$. Taking a union bound over the resulting events, we obtain that with probability $1 - \delta$, for all $n \geq 1$ and $u \in \mathcal{N}_n$,

$$f_n(u) := u^\mathsf{T} \Delta_n u + J(u)^2 \log(1/\delta_n) \geq 0.$$

Now for each $n \geq 1$, let $\pi_n \colon r\mathbb{B}_2^d \to \mathcal{N}_n$ be a map satisfying $\|u - \pi_n(u)\| \leq \epsilon_n$ for all $u \in r\mathbb{B}_2^d$. The proof will be complete once we show that for a suitable choice of $\epsilon_n$, $|f_n(u) - f_n(\pi_n(u))| \leq 5$ for all $u \in r\mathbb{B}_2^d$, and that for the chosen $\epsilon_n$, we have the bound $\log(1/\delta_n) \leq \omega_n$. We begin with the bound

$$|f_n(u) - f_n(\pi_n(u))| \leq \underbrace{|u^\mathsf{T} \Delta_n u - \pi_n(u)^\mathsf{T} \Delta_t \pi_n(u)|}_{=:A_n} + \underbrace{|J^2(u) - J^2(\pi_n(u))|}_{=:B_n} \log(1/\delta_n),$$

Letting $\|\cdot\|_{\mathrm{op}}$ denote the $\ell_2 \to \ell_2$ operator norm,

$$
\begin{aligned}
A_n &= |(u - \pi_n(u))^\mathsf{T} \Delta_n (u - \pi_n(u)) - 2\pi_n(u)^\mathsf{T} \Delta_n (\pi_n(u) - u)| \\
&\leq (\|u - \pi_n(u)\|^2 + 2\|\pi_n(u)\|\|\pi_n(u) - u\|)\|\Delta_n\|_{\mathrm{op}} \\
&\leq \epsilon_n(\epsilon_n + 2r)2n < 6\epsilon_n n. \qquad\qquad (\|\Delta_n\|_{\mathrm{op}} \leq 2n, \, r \leq 1, \, \epsilon_n < 1)
\end{aligned}
$$

Also,

$$
\begin{aligned}
B_n &= |(J(u) - J(\pi_n(u)))(J(u) + J(\pi_n(u)))| \\
&\leq 2r|(J(u) - J(\pi_n(u)))| \qquad\qquad (\forall u \in r\mathbb{B}_2^d, \, J(u) \leq \|u\| \leq r) \\
&\leq 2r|J(u - \pi_n(u))| \qquad\qquad (\forall u, u', \, |J(u) - J(u')| \leq |J(u - u')|) \\
&\leq 2r\epsilon_n < 2\epsilon_n. \qquad\qquad (r \leq 1)
\end{aligned}
$$

Now choose $\epsilon_n = 1/(4nd^2 \log(1/\delta))$. By Lemma 13, for this choice, $\log(1/\delta_n) \leq \omega_n$. Combining the bounds on $A_n$ and $B_n$, we now indeed have that for all $u \in r\mathbb{B}_2^d$,

$$|f_n(u) - f_n(\pi_n(u))| \leq A_n + B_n \log(1/\delta_n) \leq \epsilon_n(2 + 6n + \log(1/\delta_n)) \leq 5. \qquad \square$$

