# OpenReview forum: "When and why randomised exploration works (in linear bandits)"
_algorithmiclearningtheory.org/ALT/2025/Conference — ALT 2025_

### Official Review · Reviewer_aff2 · 2024-11-08
**The discussion on the assumptions**

**Rating:** 7
**Confidence:** 4

**Review:**

This work provides the nearly optimal regret upper bound $O(d\sqrt{n}log(n))$ for the (unmodified) randomised exploration algorithms (including linear TS). More specifically, their algorithm does not rely on the optimism framework, which introduces the challenge in the analysis process. To overcome this problem, this work carefully analyzes the confidence width during the algorithm process and shows that pure randomised exploration is enough to control the error terms.
Pros:
1. This work provides the first nearly optimal upper bound for the (unmodified) randomised exploration algorithms, which is meaningful.
2. The intuition of why randomised exploration algorithms work (Figure 1) is clear.

Cons:
1. Though the smooth and strongly convex assumption is stronger than previous work, I think it is meaningful to analyze the linear TS algorithm. However, it would be better to discuss the role of these two assumptions. For the strongly convex assumption, it seems to be used in Lemma 5. For the smooth assumption, it seems to be used in Eq. (2).

**Paper Award:**

No

---

### Official Review · Reviewer_xrrF · 2024-11-08
**Review comments**

**Rating:** 8
**Confidence:** 4

**Review:**

Summary:

This manuscript shows that for linear bandits satisfying certain conditions (detailed below), a randomized exploration around the ridge least square (RLS) estimate at every step achieves the optimal $\widetilde{O}(d\sqrt{n})$ regret, where $d$ is the dimension and $n$ is the time horizon. The specific algorithm is very simple: at each round $t$, let  $\widehat{\theta_t}$  be the current RLS estimate of $\theta^\star$, the learner draws a random $\theta_t = \widehat{\theta_t} + V_{t-1}^{-1/2}\eta_t$ with some noise vector $\eta_t$ (where $V_{t-1}$ is the usual information matrix in RLS up to time $t-1$) and play the optimal action associated with $\theta_t$. When $\eta_t$ is the standard Gaussian noise, the result implies that linear Thompson sampling achieves the optimal regret for linear bandits under the frequentist setup, while existing work only established a regret upper bound $\widetilde{O}(d^{3/2}\sqrt{n})$.

The key technical idea of this manuscript is to bypass the optimism-based arguments. Specifically, in the existing work [Abeille and Lazaric, 2017], it was shown that if $J(\theta_t) \ge J(\theta^\star)$ (i.e. optimism) happens with probability at least a constant (here $J(\theta)$ is the maximum reward when $\theta$ is the truth), then a UCB-style upper bound works. However, for the above lower bound to hold, the variance of $\eta_t$ must be inflated by an additional factor of $d$ which explains the additional $\sqrt{d}$ factor in the final regret. The main intuition of the current work is that, even if $J(\theta_t) \le J(\theta^\star)$, the randomized exploration due to the noise part $V_{t-1}^{-1/2}\eta_t$ will provide a sufficiently large information gain from $V_{t-1}$ to $V_t$, and mostly importantly make $V_t$ essentially balanced along all directions. This argument critically relies on the dependence of the optimal action $x_t = \nabla J(\theta_t)$ in the change of $\theta_t$, and therefore the authors need to make the following assumptions:

1. The action set is essentially continuous;
2. The function $J(\theta)^2$ is both strongly convex and smooth.

Comments:

Overall I like this manuscript. I think it is a significant result to show that linear Thompson sampling achieves the optimal frequentist regret even for linear bandits under $\ell_2$ balls, and the technical argument beyond optimism could be of independent interest. I think this paper is a great fit for ALT and I recommend acceptance.

I don't have major concerns about this manuscript, but I do have several questions and comments to discuss with the authors:

1. Do you think the consideration of $J(\theta)^2$ in Assumption 2 essential? It seems that imposing strong convexity and smoothness conditions on $J(\theta)^3$ can also make the proof go through - what do you think? A deeper understanding of this question may help identify a weaker set of conditions to make the current proof work through.

2. Do you think if it is possible to get rid of the optimism-based arguments completely? Right now the condition $J(\theta_t) \le J(\theta^\star)$ is only used at some technical steps, such as in the manner of $|a^2 - b^2| \le 2b|a-b|$ when $a\le b$. In addition, on a high level the intuition for randomized exploration should work for the case with optimism as well (up to certain technical subtleties). What do you think?

3. Right now the intuition behind Lemma 5 - Claim 7 is not well reflected in the technical analysis. I find the intuition to be simple under the $\ell_2$ ball case, but the current technical analysis is a bit lengthy and annoying. Given that some steps seem to be pretty loose (e.g. replacing $V_{t-1}$ by $V_n$ in the proof of Claim 7), I think a more direct and transparent proof should be possible and beneficial to the community.

4. The authors are encouraged to discuss the necessity of the assumptions here. I understand that the lower bounds in [Hamidi and Bayati, 2023; Zhang, 2021] for Thompson sampling imply that certain assumptions are necessary, but on the other hand the $\widetilde{O}(d^{1.5}\sqrt{n})$ upper bound in [Abeille and Lazaric, 2017] with an inflated variance does not seem to require any such conditions.

5. On Page 8, why is $\sqrt{d}$ used in the upper bound $|\xi_t|\lesssim \sqrt{d}$? I feel that even $\widetilde{O}(1)$ seems true?

6. On the last display on Page 10, although the middle term is polylog($n$), the dependence on $d$ there seems to be $d^{2.5}$. Does this mean that the final regret bound is $\widetilde{O}(d\sqrt{n} + d^{2.5})$, which is optimal only if $n \ge d^3$?

7. Page 18, before Appendix C: should $\pi$ be $M$ instead?

**Paper Award:**

No

---

> ### Author Response · Authors · 2024-11-24
>
> Dear reviewer,
>
> We thank you for taking the time to carefully read our paper! We replied regarding the assumptions in the main rebuttal. On Point 7, Page 18 is indeed a typo, thank you. For the remaining points:
>
> **On whether optimism can be removed entirely** We believe optimism is unnecessary, yet useful. In particular, our first proof for this problem, specialised for the $\ell_2$ case, involved dividing the time-steps into two regimes: steady-state, when the design matrix $V_t$ was sufficiently large, and a transitory setting before that. In bounding the regret in the steady-state setting, optimism was then not used whatsoever. In the transient regime, we showed that because $V_t$ was small, the sampling distribution was very spread out, and we had at least a constant probability of optimism. But the argument for the transient regime could be boiled down to "exploration is essentially uniform".
>
> The proof we presented instead divides steps into (A) those with a constant probability of optimism, and (B) all the other steps. With that, (A) will contain the transient regime as a subset, and may contain some of the steps in the steady-state regime. We still essentially only use optimism to handle the highly exploratory/random steps.
>
> **Page 8, $|\xi_t| \leq \sqrt{d}$ bound** There may be a tighter bound--since it is a lower-order term, we did not spend any time thinking about it. There is an easy fix, though: just assume that $J(\theta) \in [0,1]$; i.e., that $\theta \in B^d_2$. This is the standard assumption in the literature anyway. The reason we went for $\|\theta\| \leq \sqrt{d}$ is that the bounds on the Bayesian regret of Thompson sampling are with respect to $\theta_\star \sim N(0,I_d)$, and so for $\theta_\star$ with $\|\theta_\star\|\approx \sqrt{d}$, and we wanted our bound to be like-for-like comparable with the Bayesian regret bound (since our worst-case bound matches the Bayesian bound in this case).  We will change the way we present the result to depend directly on $J(\theta_\star)$.
>
> **On the $d^{5/2}$ dependence in lower order term** We opted not to include this lower-order term in the main result as it is a little complex and hard to summarise nicely. We also did not try particularly hard to get an optimal bound for it. Given your comment, we will look into this term some more, and at how to present it. In short, though, the dependence here is of the form $$\omega_n \beta_n^2 J(\theta_\star) \log(dn/(\delta \lambda)\,.$$ Now:
> 1. The $\beta_n^2 \approx d\log n$ dependence is not going anywhere.
> 2. The $\omega_n \approx d \log d$ term comes from lemma 6, the covering argument. Looking at remark 6, which gives the bound when the arm set is the $\ell_2$ ball, we see that there one can make do with $\omega_n = \log d$. Perhaps our result is loose here?
> 3. There's also a $J(\theta_\star) \leq \sqrt{d}$ dependence--we will leave this as just $J(\theta_\star)$, given our earlier discussion on assumptions on the magnitude of this.
>
> So, there's a good chance that this lower order term really is just $\widetilde O(d)$, at least for arms-sets that are not too dissimilar from $\ell_2$ balls.

---

> > ### Comment · Reviewer_xrrF · 2024-12-01
> >
> > Thank you for the detailed response! I really enjoyed reading this paper and am keeping my score.

---

### Official Review · Reviewer_Ue2r · 2024-11-08
**The paper provides conditions for randomized exploration algorithms to achieve nearly optimal regret bounds in linear bandits. This idea as well as the techniques are novel. It provides the first tight regret bound for Thompson sampling frequentist regret, although under assumptions, and limited to the no context setting.**

**Rating:** 7
**Confidence:** 3

**Review:**

The paper studies random exploration for linear bandits. Sufficient conditions for random exploration to result in nearly optimal regret are proposed; namely, smoothness and strong convexity. The class of randomized algorithms samples a $\theta = \hat{\theta}+ V \eta$, where $V$ is the least squares matrix, and $\eta$ is a randomly generated vector from a specific class of distributions.
This class of algorithms contain Thompson Sampling as a special case, hence, proving the first near optimal frequentist regret bound for Thompson sampling, albeit under extra assumptions on the action set.

Strengths:
- The paper proves the first near optimal frequentist regret bound for Thompson sampling, albeit under extra assumptions on the action set.
- Valuable insights on when randomized exploration results in nearly optimal regret are presented.
- The observation and proof that the proposed assumptions work is novel to the best of my knowledge.
- The work allows to use algorithms with better computational complexity, than those based on the optimism principle such as LinUCB, for linear bandits.

Weaknesses:
- The analysis is specific to linear bandits, and do not seem to directly extend for contextual bandits. Despite that, I believe that the papers contribution is an important step.
- It is unclear if the proposed assumptions are necessary. However, I understand that it is difficult question to answer.

Questions:
Can the work in this paper be combined with the reduction from contextual to linear bandits in [] to extend the results for contextual linear bandits? Adding a discussion on that could increase the impact of this work.


minor comments:
- please fix broken link in page 3

[1] "Contexts can be cheap: Solving stochastic contextual bandits with linear bandit algorithms." The Thirty Sixth Annual Conference on Learning Theory. PMLR, 2023.

**Paper Award:**

No

---

> ### Author Response · Authors · 2024-11-24
>
> We thank the reviewer for pointing out the work of [1]. While the lower bound construction of [2] indicates that non-inflated Thompson sampling can suffer large regret in contextual linear bandits with *adversarial* contexts, the story may be different in the *stochastic* contexts setting.
>
> Indeed, [1] proposes a reduction from contextual linear bandit operating over a sequence of arm sets $\{\mathcal{X}_1,\dots,\mathcal{X}_T\}$ randomly generated to a ficticuous linear bandit problem operating on some fixed arm set $\mathcal{X}$. As a result, any algorithm with worst-case guarantees *for all* fixed arm-set will inherit the same guarantees in the stochastic contextual setting.
>
> However, our result does not hold uniformly over arm sets, but only for those satisfying smoothness and strong-convexity.  This makes the application of [1] less clear cut: there may be problem instances where for all $t$, $\mathcal{X}_t$ satisfies our assumptions, but the resulting $\mathcal{X}$ does not. Conversely, there may be problems where $\mathcal{X}_t$ is finite (and thus does not satisfy our assumptions), but, due to randomness, $\mathcal{X}$ does.
>
> Rephrasing the approach of [1] with our notations, one can obtain $\widetilde{O}(d \sqrt{n})$ regret guarantees as long as the *average performance function* $J_{\mathcal{X}}(\theta) = \mathbb{E}_{X_t \sim \mathcal{D} } \left( J_{\mathcal{X}_t} (\theta) \right)$ is suitably smooth and convex, where $J_{\mathcal{X}_t} (\theta) = \max_{x\in\mathcal{X}_t} x^\top \theta$ is the per-step optimal performance (which may not be smooth and strongly convex).
>
> We will happily add a discussion of this work to our manuscript.
>
> [2] : N. Hamidi and M. Bayati. On the frequentist regret of linear Thompson sampling

---

> > ### Comment · Reviewer_Ue2r · 2024-11-27
> >
> > Thank you for your detailed response.

---

### Author Rebuttal · Authors · 2024-11-24

The main question raised by the reviewers was on our choice of assumptions. We address this here.

**On what the assumptions should be** Recall the two lower bounds we cite. The construction in both is to pick an arm-set such that for an exponentially large proportion of directions, $0$ is the optimal arm--which requires that $0$ is on the boundary of the arm-set. We conjecture that a sufficient assumption for Thompson sampling to work (without inflation) is that there exists a ball of constant radius around the origin that does not intersect with the arm-set. That is, $0$ is either within or without the arm set, and is strictly separated from it. Of course, one could further relax this to it only holding for most directions--a couple directions where $0$ is optimal should not break things. This intuition comes from trying to find other ways to break the algorithm, and always ending back up at making the algorithm get stuck selecting arm $0$ for $e^d$-many time-steps in the early regime.

Our proof makes the stronger assumption that $0$ is in the interior of the arm set, and disallows it being on the (strict) exterior (which we think should be fine too). Why is this? Well, we started with a proof for the $\ell_2$ ball, and then generalised it. The $\ell_2$ ball case already had the assumption that $0$ is on the interior built into it, and so our proof techniques were biased toward that assumption and hard to generalise to the other case.

The strong convexity and smoothness assumptions are clearly not necessary: Thompson sampling works well for the multi-armed setting, after all. Interestingly, we looked at the Thompson sampling proof for the multi-armed setting from the Bandit Algorithms book (Szepesvari & Lattimore), and there are very direct parallels between steps in that proof and steps in ours. We hope to find a more general proof in the future.

**Reviewer xrrf noted that inflated TS needs no extra assumptions** Indeed, the lower bounds stop working in a very clear way once $\sqrt{d}$ inflation is added, but it is exactly $\sqrt{d}$ inflation that is needed.

---

> ### Comment · Reviewer_aff2 · 2024-12-02
>
> Thanks for the detailed discussion on the assumptions. I am happy to maintain my score.

---

### Meta-Review · Area_Chair_wXtj · 2024-12-01

**Recommendation:** Accept
**Confidence:** 4

**Metareview:**

This paper considers the frequentist behavior of randomized exploration algorithms for linear bandits (in which we perturb the parameter estimate at each iteration, and then compute the greedy action). Previous work studying this problem requires a certain "inflation" of the posterior, which leads to suboptimal dependence on dimension, and it is known that various randomized exploration schemes can fail without this type of intervention. The present work makes new progress on this problem by showing that whenever the action space satisfies certain smoothness and strong convexity properties (generalizing the l_2 ball), randomized exploration can achieve optimal dependence on dimension.

Reviewers were generally quite positive on this paper, and found the results to be interesting and technically novel, making progress on an important problem in the literature on bandits. They also found it to be well written. Overall, I think it is a good fit for ALT, and recommend acceptance.

In the discussion period, there were many interesting points raised by the reviewers and authors regarding the necessity and intuition behind specific assumptions. I encourage the authors to incorporate these points into the final revision of the paper.

**Paper Award:**

No